# Germinated Chickpea and Lupin as Promising Ingredients for Breadmaking—Rheological Features

**Denisa Atudorei** [1], **Mădălina Ungureanu-Iuga** [1,2], **Georgiana Gabriela Codină** [1,*] **and Silvia Mironeasa** [1]

1 Faculty of Food Engineering, Ştefan cel Mare University of Suceava, 13 Universitatii Street, 720229 Suceava, Romania; denisa.atudorei@outlook.com (D.A.); madalina.iuga@usm.ro (M.U.-I.); silviam@fia.usv.ro (S.M.)
2 Integrated Center for Research, Development and Innovation in Advanced Materials, Nanotechnologies, and Distributed Systems for Fabrication and Control (MANSiD), Ştefan cel Mare University of Suceava, 13th University Street, 720229 Suceava, Romania
* Correspondence: codina@fia.usv.ro

**Abstract:** Improving the alpha-amylase activity of wheat flour represents an opportunity to valorize wheat grains of low baking quality. In this sense, germinated legumes can be used to increase enzymatic activity, giving superior final product characteristics at the same time. The aim of this study was to underline the effects of chickpea (CGF) and lupin germinated flours (LGF) added simultaneously to white wheat flour on the rheological behavior of dough and to evaluate an optimal product microstructure. For this purpose, the falling number, dough rheological properties during mixing, 3D-deformation and fermentation, and the visco-elastic behavior were evaluated, the effects of factors (CGF and LGF levels) and their optimization have been studied by applying a full factorial design and response surface methodology (RSM). The LGF sample had a composition of 39.4% protein, 10.3% moisture, 6.9% fat, and 3.4% ash, whereas the CGF presented 21.1 % protein, 9.4% moisture, 5.2% fat, and 3.6% ash. The results showed that CGF and LGF determined the decrease of the falling number, dough water absorption, tolerance to kneading, dough consistency at 250 and 450 s, extensibility, the maximum height of the gas release curve, volume of gas retained by the dough at the end of the test, total volume of $CO_2$ production, visco-elastic moduli, and gelatinization temperatures. On the other hand, dough elasticity and alveograph curve ratio increased proportionally to the increase of CGF and LGF addition levels. The optimal combination considering the rheological properties of dough was found to be 8.57% CGF, 5.31% LGF, and 86.12% wheat flour, with enhanced alpha-amylase activity being obtained compared to the control. These results provide valuable information on the possibility of using germinated legumes such as chickpeas and lupin in breadmaking to enhance wheat flour technological properties (besides traditionally used barley malt flour).

**Keywords:** germination; legumes; rheology; white wheat flour; dough microstructure

## 1. Introduction

During the last few decades, consumers have increasingly been concerned about a healthy diet, demanding food products with a high nutritional value, good sensory characteristics, and low prices. Unfortunately, many food producers are focused on improving the sensory characteristic of food products often to the detriment of their nutritional value. This fact is also evident in the case of the bakery industry where the most consumed products are those obtained from refined wheat flour. Through refining, wheat flour loses valuable nutritional substances such as minerals, vitamins, and fibers (since they are found mainly in the outer layers of the grain shell) [1]. Moreover, to improve the sensory characteristics of bakery products, to shorten the manufacturing technological process, and to minimize production costs, different chemical additives are used. Most of them may have different adverse health effects in time, with some of them presenting carcinogenic potential (e.g.,

potassium bromate used in some countries from the United States of America [2]). Then, the question arises as to "how these shortcomings that today's consumer is facing could be alleviated?". One solution is the use of germinated legumes in the bakery industry which may limit the necessity of different additives, with a positive effect from a technological and nutritional point of view. Through germination, the enzymatic activity of the grains and seeds subjected to this process is activated and the bioavailability of nutrients in the grain is increased [3]. For example, the increase of vitamins and minerals amounts could be explained by the activity of phytase that split the link between minerals, enzymes, and proteins, which led to the release of vitamins and minerals in the germination stage [4,5]. Therefore, through germination, the amount of phytic acid in the grains decreases, and the amount of minerals and vitamins increases [6]. Another advantage of the germination process, with prospects for applications in bakeries, is that during the germination process there is an activation of its endoenzymes. This makes possible the use of sprouted flours as an improver ingredient for bakery products obtained from wheat flour of low alpha-amylase activity (and therefore with a high falling number index value) [7].

Chickpea and lupin are legumes that can be successfully used as ingredients in bread-making recipes to correct their nutritional deficiencies. Lupin and chickpea contain a significant amount of protein [8,9] and have a balanced amino acid profile. Lupin is composed of all essential amino acids and a wide range of minerals [10]. Lupin and chickpea are important sources of dietary fibers [11] and contain significant amounts of phenolic compounds [9,12]. The quantity of protein found in lupin is higher than in other legumes [13], being rich in lysine, arginine [14], and asparagine [15]. In turn, chickpea contains a significant amount of zinc, calcium, magnesium, and iron (4.1 mg, 160 mg, 138 mg, and 5.0 mg per 100 g of chickpea) [9]. Different studies have shown that isoflavones from chickpea (trifolyrhizine, genistein, formononetin, sissotrin) have beneficial effects in treating some medical problems such as hyperlipidemia, osteoporosis, cardiovascular disease, cancer [16]. Lupin consumption could prevent different diseases such as diabetes, hyperlipidemia, and hypertension [17].

Chickpea occupies third place in terms of legume consumption in the world, being eco-friendly and sustainable for the environment. It has also an important role in fixing atmospheric nitrogen in the soil [18]. Also, lupin cultivation does not require special conditions, it has a relatively high tolerance to abiotic stressors, and a potential to restore poor or contaminating soils [14]. However, an important aspect related to lupin grains is the content of alkaloids. The existing varieties of lupin contain different amounts of alkaloids. There are different techniques to reduce the amount of alkaloids in lupin grains, but there are also varieties that are not considered toxic since they have an alkaloid content that does not exceed the recommended limit. This category includes the sweet lupin variety. From the multitude of lupin varieties (more than 170), those with a very low alkaloid content (less than 0.02%) are the white variety (*Lupinus albus* L.), the yellow variety (*Lupinus luteus* L.), and the narrow-leaved lupins (*Lupinus angustifolius* L.) [14]. These species of lupin are known as sweet lupin [19].

Various studies have shown the possibility of using sweet lupin and chickpea flour as ingredients to the recipe for bread making of white wheat flour in order to improve its nutritional quality, without adversely affecting its technological properties. Wandersleben et al. [11] highlighted that the addition of a maximum of 10% lupin flour did not negatively affect bread quality. In the case of lupin flour addition (maximum 10%), the bread obtained had a higher volume due to the improvement of dough capacity to retain the carbon dioxide resulting in the leavening and baking stage. In a study by Kotsiou et al. [20], it was concluded that the addition of a maximum of 10% roasted chickpea flour in wheat flour did not negatively affect dough rheological properties and bread quality. Furthermore, Mohammed et al. [21] highlighted the fact that an addition of 10–20% chickpea flour can be successfully used in the bread-making recipe. Yousif and Safaa [22] reported higher nutritional value of gluten free bread enriched with germinated chickpea and sweet lupin, while the positive effects of germination were observed by the decrease of

antinutrients contents. Similar increases in nutrients were obtained for Arabic flat bread when germinated lupin was added [23].

The aim of this study was to highlight the optimal amount of germinated chickpea and white lupin flour that can be added in a germinated form in bread making to obtain optimum dough behavior during mixing, 3D-deformation, pasting, and fermentation. The importance of this study also derives from the fact that the possibility of using the addition of germinated chickpea flour and germinated white lupin flour together in the bread-making recipe has not been studied so far.

## 2. Materials and Methods

### 2.1. Materials

In this study, white wheat flour (2020 harvest) provided by S.C. Dizing S.R.L. company (Brusturi, Neamt, Romania) was used. Lupin (*Lupinus albus*) and chickpea (*Cicer arietinium* L.) were used for wheat flour substitution, both types of legumes grains being subjected to germination for four days, lyophilization, and milling according to the protocol reported in our previous studies [7,24]. Germinated legumes were milled in a laboratory mill 3100 (Perten Instruments, Hägersten, Sweden) and the flours obtained had particle sizes <180 μm.

All flours types were analyzed according to the Romanian or international standards methods as following: moisture content (ICC 110/1), protein content (ICC 105/2), wet gluten content (SR 90:2007), fat content (ICC 136), ash content (ICC 104/1), and gluten deformation index (SR 90:2007). Gluten deformation index method is based on the relation between gluten quality and its deformation capacity when it is left at rest, the gluten deforming all the more so as its quality is poorer [25]. In order to test germinated lupin (LGF) and chickpea (CGF) flours safety, they were analyzed for mycotoxins by using an ELISA kit (Prognosis Biotech, Larissa, Greece), different microorganisms such as *Bacillus cereus* (SR EN ISO 7932-2003:2005), and for total yeast and molds (SR ISO 7954:2001).

### 2.2. Dough Rheology

#### 2.2.1. Empirical Dough Rheological Properties

Empirical dough rheological properties during mixing and 3D-deformation were determined by using an Alveo Consistograph device (Chopin Technologies, Cedex, France). The mixing properties in terms of adapted water absorption capacity (WA), tolerance to kneading (Tol), consistency of the dough after 250 s (D250) and 450 s (D450) were determined with the Consistograph part according to ICC 171 method, while the 3D-deformation properties such as maximum pressure (P), dough extensibility (L), alveograph energy (W) and alveograph curve ratio (P/L) were determined with the Alveograph part according to ICC 121 method. The data were analyzed with AlpcWin software, version 3.14 (Chopin Technology).

Flour mixes falling number values were determined by using the falling number device (FN 1305, Perten Instruments AB, Stockholm, Sweden).

The rheological characteristics of dough during fermentation of the flour mixes were determined with the Rheofermentometer device (Chopin Rheo, type F3, Villeneuve-La-Garenne Cedex, France). Dough fermentation properties in terms of maximum height of the gas release curve (H'm, mm), total volume of $CO_2$ production (VT, mL), the volume of gas retained in the dough at the end of the test (VR, mL), and retention coefficient (CR, %) were determined according to AACC89–01.01 method by using a dough obtained from 250 g mix flour, 7 g of compressed yeast (*Saccharomyces cerevisiae*) 5 g of salt and distilled water according to the Consistograph adapted water absorption value, mixed in the Alveo Consistograph device for 3–6 min.

2.2.2. Fundamental Dough Rheological Properties

Fundamental dough rheological properties analysis was made by using a HAAKE MARS 40 rheometer (Thermo-HAAKE, Karlsruhe, Germany). The dough samples were prepared in the Consistograph mixer according to the water absorption value of the flour mix, then they were rested for 30 min before being analyzed. The storage modulus ($G'$), loss modulus ($G''$), and loss tangent (tan δ) were determined by frequency sweep tests and the values were considered at 1 Hz. For the temperature sweep test in which dough samples were heated from 25 °C to 100 °C at a rate of 4 °C per min, the initial ($T_i$) and maximum ($T_{max}$) gelatinization temperatures were determined.

*2.3. Dough Microstructure*

Dough microstructures of the optimal sample with germinated chickpea and lupin added in wheat flour and of the control were analyzed by epifluorescence light microscopy (EFLM) with a Motic AE 31 (Motic, Optic Industrial Group, Xiamen, PR China) device with catadioptric objectives LWD PH 203 (N.A. 0.4). Dough samples conditioning and images processing were made according to the protocols previously described in our studies [26,27].

*2.4. Data Statistical Analysis*

Data modeling was carried out on the trial version of Design Expert software (Stat-Ease, Inc., Minneapolis, MN, USA). The values of the responses are presented in Table 1a,b.

The influence of LGF and CGF on dough rheological properties were evaluated by using Response Surface Methodology (RSM) and a full factorial design with two factors (A—CGF and B—LGF amounts), both varied at 0, 5, 10, 15, and 20% as a flour substitute. The best predictive model for the experimental data variation for each response was chosen by taking into account F-test values, coefficient of determination ($R^2$), and adjusted coefficients of determination (*Adj.-$R^2$*). Analysis of variance (ANOVA) was applied in order to evaluate the influence of factors and their interactions (significant at $p < 0.05$) on the following responses: FN (falling number), WA (water absorption), Tol (tolerance to kneading), D250/D450 (dough consistency after 250 and 450 s, respectively), P (dough elasticity), L (dough extensibility), W (alveograph energy), P/L (alveograph curve ratio), $H'm$ (maximum height of the gas release curve), VT (total volume of $CO_2$ production), VR (the volume of the gas retained in the dough at the end of the test), CR (retention coefficient), $G'$ (elastic modulus), $G''$ (viscous modulus), tan δ (loss tangent), $T_i$ (initial gelatinization temperature), and $T_{max}$ (maximum gelatinization temperature).

The optimization of factors was carried out by applying the desirability function. For this purpose, the factors (CGF and LGF amounts) were kept in range, the Falling number parameter was minimized, the rheological parameters during fermentation were maximized, while the rheological characteristics during mixing and 3D-deformation, and the fundamental rheological parameters were kept in range. The loss tangent was excluded from the optimization since the mathematical model was non-significant ($p > 0.05$). The experimental design and the optimization were done on the trial version of Design Expert software (Stat-Ease, Inc., Minneapolis, MN, USA).

*Student-t*-test was employed to evaluate the differences considered significant at $p < 0.05$ between the optimal and control sample. Correlations, principal component analysis and statistical tests were performed on XLSTAT software for Excel 2021 version (Addinsoft, New York, NY, USA).

**Table 1.** (**a**). Effects of CGF and LGF on falling number and dough consistograph and alveograph parameters. (**b**). Effects of CGF and LGF on dough properties during fermentation and dynamic rheology.

| (a) | | | | | | | | | |
|---|---|---|---|---|---|---|---|---|---|
| | **Responses** | | | | | | | | |
| **Run** | **Falling Number** | **Consistograph** | | | | **Alveograph** | | | |
| | **FN (s)** | **WA (%)** | **Tol (s)** | **D250 (mbar)** | **D450 (mbar)** | **P (mm)** | **L (mm)** | **W (10⁻⁴ J)** | **P/L (adim.)** |
| 1 | 247 ± 4.24 | 52.1 ± 0.14 | 126 ± 4.24 | 163 ± 4.24 | 569 ± 2.83 | 133 ± 1.41 | 29 ± 1.41 | 164 ± 2.83 | 4.59 ± 2.83 |
| 2 | 277 ± 2.83 | 52.6 ± 0.14 | 149 ± 4.24 | 174 ± 2.83 | 624 ± 5.66 | 127 ± 1.41 | 33 ± 2.83 | 174 ± 4.24 | 3.85 ± 0.29 |
| 3 | 206 ± 2.83 | 50.5 ± 0.07 | 106 ± 4.24 | 123 ± 4.24 | 597 ± 5.66 | 143 ± 1.41 | 12 ± 0.83 | 80 ± 2.83 | 11.91 ± 0.78 |
| 4 | 254 ± 1.41 | 52.2 ± 0.00 | 194 ± 7.07 | 204 ± 2.83 | 731 ± 2.83 | 136 ± 2.83 | 25 ± 1.41 | 153 ± 2.83 | 5.44 ± 0.42 |
| 5 | 268 ± 1.41 | 52.5 ± 0.28 | 181 ± 1.41 | 214 ± 2.83 | 785 ± 4.24 | 123 ± 2.83 | 35 ± 2.83 | 172 ± 4.24 | 3.51 ± 0.21 |
| 6 | 283 ± 2.83 | 52.7 ± 0.14 | 209 ± 2.83 | 218 ± 1.41 | 754 ± 1.41 | 110 ± 2.83 | 39 ± 1.41 | 172 ± 5.66 | 2.82 ± 0.17 |
| 7 | 255 ± 2.83 | 52.2 ± 0.14 | 163 ± 4.24 | 194 ± 1.41 | 761 ± 2.83 | 134 ± 1.41 | 28 ± 1.41 | 168 ± 1.41 | 4.79 ± 0.30 |
| 8 | 234 ± 1.41 | 52.4 ± 0.14 | 138 ± 2.83 | 176 ± 2.83 | 623 ± 2.83 | 142 ± 1.41 | 22 ± 2.83 | 159 ± 4.24 | 6.45 ± 0.78 |
| 9 | 302 ± 2.83 | 53.2 ± 0.00 | 167 ± 1.41 | 185.5 ± 3.54 | 762.5 ± 6.36 | 122 ± 2.83 | 40.5 ± 3.54 | 198 ± 4.24 | 3.02 ± 0.19 |
| 10 | 228 ± 2.83 | 51.7 ± 0.28 | 191 ± 1.41 | 223 ± 2.83 | 834 ± 2.83 | 149 ± 2.83 | 21 ± 1.41 | 137 ± 2.83 | 7.10 ± 0.35 |
| 11 | 279 ± 1.41 | 52.5 ± 0.14 | 201 ± 2.83 | 205 ± 1.41 | 732 ± 4.24 | 130 ± 1.41 | 33 ± 2.83 | 180 ± 1.41 | 3.94 ± 0.05 |
| 12 | 350 ± 2.83 | 54.3 ± 0.14 | 214 ± 2.83 | 394 ± 1.41 | 943 ± 4.24 | 104 ± 1.41 | 72 ± 2.83 | 301 ± 4.24 | 1.44 ± 0.04 |
| 13 | 249 ± 1.41 | 51.9 ± 0.14 | 188 ± 1.41 | 217 ± 2.83 | 833 ± 4.24 | 146 ± 2.83 | 20 ± 1.41 | 126 ± 2.83 | 7.30 ± 0.66 |
| 14 | 211 ± 1.41 | 51.2 ± 0.28 | 131 ± 2.83 | 168 ± 2.83 | 617 ± 2.83 | 150 ± 1.41 | 16 ± 1.41 | 113 ± 2.83 | 9.37 ± 0.92 |
| 15 | 260.5 ± 2.12 | 52.3 ± 0.28 | 200.5 ± 6.36 | 242 ± 4.24 | 881 ± 5.66 | 107 ± 1.41 | 38 ± 2.83 | 156 ± 4.24 | 2.83 ± 0.25 |
| 16 | 318 ± 5.66 | 53.6 ± 0.00 | 215 ± 2.83 | 225 ± 4.24 | 785 ± 5.66 | 106 ± 1.41 | 46 ± 2.83 | 190 ± 4.24 | 2.30 ± 0.11 |
| 17 | 236 ± 4.24 | 51.8 ± 0.14 | 118 ± 2.83 | 155 ± 1.41 | 537 ± 2.83 | 144 ± 2.83 | 18 ± 1.41 | 127 ± 4.24 | 8.00 ± 0.47 |
| 18 | 203 ± 1.41 | 50.7 ± 0.14 | 111 ± 1.41 | 128 ± 2.83 | 614 ± 2.83 | 151 ± 2.83 | 12 ± 1.41 | 75 ± 2.83 | 12.58 ± 1.73 |
| 19 | 320 ± 2.83 | 53.7 ± 0.28 | 181 ± 2.83 | 339 ± 5.66 | 878 ± 2.83 | 113 ± 0.00 | 46 ± 1.41 | 204 ± 4.24 | 2.46 ± 0.08 |
| 20 | 194 ± 4.24 | 50.3 ± 0.28 | 101 ± 1.41 | 118 ± 2.83 | 621 ± 4.24 | 158 ± 1.41 | 10 ± 1.41 | 64 ± 2.83 | 15.80 ± 2.35 |
| 21 | 236 ± 1.41 | 51.6 ± 0.14 | 177 ± 4.24 | 216 ± 1.41 | 713 ± 2.83 | 123 ± 1.41 | 16 ± 1.41 | 91 ± 1.41 | 7.69 ± 0.59 |
| 22 | 308 ± 2.83 | 53.8 ± 0.28 | 204 ± 2.83 | 317 ± 2.12 | 903 ± 4.24 | 114 ± 2.83 | 46 ± 1.41 | 193 ± 2.83 | 2.52 ± 0.01 |
| 23 | 218 ± 1.41 | 50.9 ± 0.14 | 117 ± 2.83 | 133 ± 4.24 | 632 ± 4.24 | 128 ± 2.83 | 19 ± 1.41 | 113 ± 1.41 | 6.74 ± 0.35 |
| 24 | 242 ± 4.24 | 51.8 ± 0.14 | 184 ± 1.41 | 231 ± 5.66 | 729 ± 4.24 | 125 ± 1.41 | 23 ± 1.41 | 132 ± 2.83 | 5.43 ± 0.27 |
| 25 | 329 ± 2.83 | 53.9 ± 0.14 | 223 ± 2.83 | 289 ± 7.07 | 931 ± 5.66 | 108 ± 1.41 | 55 ± 2.83 | 215 ± 2.83 | 1.96 ± 0.13 |

| (b) | | | | | | | | | |
|---|---|---|---|---|---|---|---|---|---|
| | **Responses** | | | | | | | | |
| **Run** | **Rheofermentometer** | | | | **Rheometer** | | | | |
| | **H'm (mL)** | **VT (mL)** | **VR (mL)** | **CR (%)** | **G' (Pa)** | **G'' (Pa)** | **tan δ (adim.)** | **Ti (·C)** | **Tmax (°C)** |
| 1 | 64.3 ± 0.28 | 1425 ± 4.24 | 1147 ± 4.24 | 80.5 ± 0.00 | 13,440 ± 2.83 | 6235 ± 2.83 | 0.4630 ± 0.00 | 48.8 ± 0.00 | 74.8 ± 0.14 |
| 2 | 68.0 ± 0.00 | 1550 ± 2.83 | 1159 ± 4.24 | 74.8 ± 0.21 | 25,460 ± 1.41 | 10,050 ± 2.83 | 0.3790 ± 0.00 | 49.3 ± 0.14 | 73.3 ± 0.14 |
| 3 | 46.2 ± 0.28 | 1093 ± 4.24 | 947 ± 2.83 | 86.6 ± 0.14 | 19,700 ± 4.24 | 7952 ± 4.24 | 0.4037 ± 0.00 | 49.1 ± 0.14 | 72.9 ± 0.28 |
| 4 | 62.3 ± 0.14 | 1439 ± 8.49 | 1207 ± 2.83 | 83.9 ± 0.71 | 42,678 ± 2.12 | 16,233 ± 4.24 | 0.3804 ± 0.00 | 50.9 ± 0.28 | 73.8 ± 0.14 |
| 5 | 69.7 ± 0.42 | 1493 ± 5.66 | 1177 ± 4.24 | 78.8 ± 0.00 | 37,781 ± 1.41 | 14,921 ± 2.83 | 0.3949 ± 0.00 | 51.0 ± 0.14 | 74.4 ± 0.14 |
| 6 | 66.3 ± 0.28 | 1641 ± 4.24 | 1233 ± 2.83 | 75.2 ± 0.14 | 46,600 ± 4.24 | 16,140.5 ± 3.54 | 0.3460 ± 0.00 | 50.5 ± 0.00 | 74.8 ± 0.14 |
| 7 | 57.8 ± 0.28 | 1344 ± 4.24 | 1118 ± 4.24 | 83.2 ± 0.00 | 23,065 ± 2.83 | 8932 ± 5.66 | 0.3873 ± 0.00 | 49.5 ± 0.14 | 74.8 ± 0.28 |
| 8 | 57.6 ± 0.14 | 1281 ± 2.83 | 1002 ± 2.83 | 78.2 ± 0.42 | 19,873 ± 4.24 | 8137 ± 4.24 | 0.4095 ± 0.00 | 49.9 ± 0.28 | 75.0 ± 0.14 |
| 9 | 70.5 ± 0.14 | 1621 ± 5.66 | 1214 ± 2.83 | 74.9 ± 0.14 | 43,240 ± 2.83 | 16,060 ± 4.24 | 0.3710 ± 0.00 | 50.4 ± 0.14 | 73.9 ± 0.14 |
| 10 | 55.8 ± 0.14 | 1307 ± 2.83 | 1056 ± 4.24 | 80.7 ± 0.28 | 28,762 ± 4.24 | 10,167 ± 4.24 | 0.3535 ± 0.00 | 49.8 ± 0.14 | 73.6 ± 0.28 |
| 11 | 66.8 ± 0.14 | 1481 ± 5.66 | 1142 ± 2.83 | 77.1 ± 0.14 | 43,446 ± 2.83 | 16,578 ± 2.83 | 0.3816 ± 0.00 | 49.8 ± 0.14 | 74.7 ± 0.28 |
| 12 | 65.9 ± 0.14 | 1532 ± 4.24 | 1228 ± 2.83 | 80.2 ± 0.07 | 29,290 ± 5.66 | 10,780 ± 4.24 | 0.3680 ± 0.00 | 51.9 ± 0.14 | 73.4 ± 0.28 |
| 13 | 55.2 ± 0.28 | 1366 ± 2.83 | 1037 ± 2.83 | 75.9 ± 0.00 | 28,122 ± 4.24 | 10,243 ± 2.83 | 0.3642 ± 0.00 | 50.2 ± 0.14 | 74.7 ± 0.14 |
| 14 | 44.0 ± 0.42 | 1071 ± 2.83 | 916 ± 4.24 | 85.5 ± 0.14 | 26,211 ± 2.83 | 9085 ± 4.24 | 0.3466 ± 0.00 | 49.5 ± 0.28 | 73.3 ± 0.14 |
| 15 | 63.1 ± 0.00 | 1394 ± 5.66 | 1148 ± 4.24 | 82.4 ± 0.64 | 30,730 ± 1.41 | 11,630 ± 2.83 | 0.3780 ± 0.00 | 49.8 ± 0.00 | 75.1 ± 0.14 |
| 16 | 72.2 ± 0.14 | 1762 ± 7.07 | 1208.5 ± 4.95 | 68.5 ± 0.00 | 41,130 ± 2.83 | 14,230 ± 5.66 | 0.3450 ± 0.00 | 51.2 ± 0.14 | 74.5 ± 0.14 |
| 17 | 46.4 ± 0.00 | 1051 ± 0.71 | 928 ± 4.24 | 88.2 ± 0.57 | 28,648 ± 2.83 | 9843 ± 4.24 | 0.3436 ± 0.00 | 49.5 ± 0.14 | 74.9 ± 0.14 |
| 18 | 51.2 ± 0.14 | 1300 ± 2.83 | 1004 ± 5.66 | 77.3 ± 0.71 | 22,146 ± 4.24 | 8345 ± 2.83 | 0.3768 ± 0.00 | 48.7 ± 0.00 | 74.3 ± 0.28 |
| 19 | 74.5 ± 0.14 | 1651 ± 4.24 | 1239 ± 2.83 | 75.1 ± 0.35 | 34,400 ± 5.66 | 13,000 ± 2.83 | 0.3770 ± 0.00 | 51.1 ± 0.14 | 73.5 ± 0.14 |
| 20 | 38.1 ± 0.28 | 974 ± 4.24 | 802 ± 2.83 | 82.3 ± 0.71 | 19,415 ± 1.41 | 6713 ± 4.24 | 0.3458 ± 0.00 | 48.4 ± 0.28 | 73.8 ± 0.28 |
| 21 | 61.0 ± 0.28 | 1340 ± 5.66 | 1117 ± 2.83 | 83.4 ± 0.28 | 25,742 ± 2.83 | 9367 ± 2.83 | 0.3639 ± 0.00 | 49.0 ± 0.14 | 74.6 ± 0.00 |
| 22 | 79.6 ± 0.28 | 1763 ± 2.83 | 1614 ± 4.24 | 91.5 ± 0.14 | 42,185 ± 4.24 | 15,145 ± 5.66 | 0.3590 ± 0.00 | 50.8 ± 0.14 | 74.2 ± 0.14 |
| 23 | 46.4 ± 0.14 | 1164 ± 1.41 | 960 ± 5.66 | 82.5 ± 0.28 | 20,847 ± 4.24 | 8004 ± 5.66 | 0.3839 ± 0.00 | 49.1 ± 0.14 | 75.3 ± 0.14 |
| 24 | 62.9 ± 0.14 | 1282 ± 2.83 | 1187 ± 2.12 | 92.6 ± 0.14 | 29,165 ± 4.24 | 11,145 ± 2.83 | 0.3821 ± 0.00 | 49.3 ± 0.03 | 74.9 ± 0.14 |
| 25 | 71.3 ± 0.28 | 1622 ± 8.49 | 1260 ± 4.24 | 77.7 ± 0.14 | 32,300 ± 7.07 | 12,250 ± 4.24 | 0.3790 ± 0.00 | 51.6 ± 0.00 | 73.7 ± 0.28 |

(a) FN, falling number; WA, water absorption; Tol, tolerance to kneading; D250/D450, dough consistency after 250 and 450 s, respectively; P, dough elasticity; L, dough extensibility; W, alveograph energy; P/L, alveograph curve ratio. (b) H'm, maximum height of the gas release curve; VT, total volume of $CO_2$ production; VR, the volume of the gas retained in the dough at the end of the test; CR, retention coefficient; G', G'', elastic and viscous modulus; tan δ, loss tangent; $T_i$, initial gelatinization temperature; $T_{max}$, maximum gelatinization temperature.

## 3. Results

### 3.1. Materials Characteristics

The chemical characteristics of the white wheat flour used as the base material in this study were: 0.66% ash content, 14.6% moisture content, 12.3% protein content, 1.12% fat content, 30.4% wet gluten content, 3 mm gluten deformation index, a falling number value of 356 s, alveograph energy of $301 \times 10^{-4}$ J, and P/L of 1.43. Gluten deformation index less than 6 mm indicate a tenacious gluten, while values higher than 20 mm indicate a weak,

sticky gluten characterized by a very fast proteolytic degradation process [28]. According to the data obtained the wheat flour has a low alpha-amylase activity and is of a strong quality for bread making [29]. The germinated lupin (LGF) presented 39.4% protein, 10.3% moisture, 6.9% fat, and 3.4% ash, whereas the germinated chickpea (CGF) presented 21.1% protein, 9.4% moisture, 5.2% fat, and 3.6% ash. From a microbiological point of view, the germinated flour samples presented a total amount of 1 UFC/g yeast and molds and were free of *Bacillus cereus*. Mycotoxins values for LGF and CGF were the following: for aflatoxin less than 1.4 ppb, for ochratoxin of 9.71 ppb and 19.08 ppb and zearalenone of 45.70 and 47.25 ppb, respectively. According to the data obtained the LGF and CGF may be used as ingredients in bread making [30,31].

### 3.2. Influence of CGF and LGF on Falling Number

The incorporation of CGF and LGF in wheat flour caused significant changes in flour falling number and dough mixing and 3D-deformation rheological properties. Falling number, dough elasticity, and extensibility data were successfully fitted to the quadratic model which described between 81 and 98% of the variation (Table 2).

**Table 2.** ANOVA results of the models fitted for falling number and dough rheological properties (consistograph and alveograph ones).

| Factors | Parameters | | | | | | | | |
|---|---|---|---|---|---|---|---|---|---|
| | FN (s) | WA (%) | Tol (s) | D250 (mbar) | D450 (mbar) | P (mm) | L (mm) | W ($10^{-4}$ J) | P/L (adim.) |
| Constant | 254.23 | 52.26 | 167.60 | 201.95 | 735.60 | 135.64 | 25.20 | 154.00 | 5.75 |
| A | −37.12 *** | −0.96 *** | −8.36 * | −38.57 | −35.76 | 7.32 ** | −12.20 *** | −48.96 *** | 2.85 *** |
| B | −44.36 *** | −1.10 *** | −49.40 *** | −66.19 * | −133.52 *** | 17.60 *** | −15.76 *** | −47.48 *** | 3.65 *** |
| A × B | 8.00 ** | 0.06 | −6.92 | 27.76 * | 46.56 | 3.52 | 4.04 ** | 5.80 | 2.25 *** |
| $A^2$ | 4.80 | | | 22.06 | | −4.40 | 3.26 * | | |
| $B^2$ | 7.37 * | | | −5.89 | | −8.80 * | 6.74 *** | | |
| $A^2B$ | | | | −53.49 * | | | | | |
| $AB^2$ | | | | −38.06 | | | | | |
| $A^3$ | | | | 25.60 | | | | | |
| $B^3$ | | | | 25.33 | | | | | |
| Model evaluation | | | | | | | | | |
| $R^2$ | 0.98 | 0.95 | 0.88 | 0.89 | 0.73 | 0.81 | 0.96 | 0.91 | 0.94 |
| *Adj.-*$R^2$ | 0.97 | 0.94 | 0.86 | 0.83 | 0.70 | 0.76 | 0.95 | 0.90 | 0.93 |
| *p*-value | <0.0001 | <0.0001 | <0.0001 | <0.0001 | <0.0001 | <0.0001 | <0.0001 | <0.0001 | <0.0001 |

*** $p < 0.001$, ** $p < 0.01$, * $p < 0.05$, A: CGF—chickpea germinated flour addition (%), B: LGF, lupin germinated flour addition (%), $R^2$, *Adj.-*$R^2$, measures of model fit; FN, falling number; WA, water absorption; Tol, tolerance to kneading; D250/D450, dough consistency after 250 and 450 s, respectively; P, dough elasticity; L, dough extensibility; W, alveograph energy; P/L, alveograph curve ratio.

Falling number is a measure of the alpha-amylase activity of flour. The addition of CGF and LGF in wheat flour resulted in a decrease of falling number values as the amount was higher (Figure 1), with both factors and their interaction being significant ($p < 0.05$).

### 3.3. Influence of CGF and LGF on Dough Rheological Behavior during Mixing and 3D-Deformation

Water absorption, tolerance to kneading, dough consistency after 450 s, alveograph energy, and alveograph curve ratio values were fitted to the 2FI model which described between 73 and 95% of data variation, while the cubic model explained 89% of dough consistency after 250 s values variations. All of the mathematical models listed above were significant at $p < 0.0001$ (Table 2).

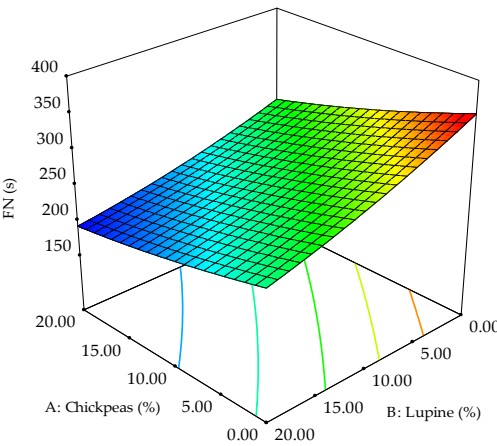

**Figure 1.** Response surface plot of falling number (FN) as a function of chickpea germinated flour (CGF) and lupin germinated flour (LGF) addition.

The rheological behavior of dough during mixing depends on the structure and interactions between dough matrix components, the incorporation of CGF and LGF leading to lower water absorption and kneading tolerance of dough as the addition level increased (Figure 2). Since the models used are first-order, the fitted response surfaces are plane. The CGF and LGF factors exerted significant ($p < 0.05$) influence on these parameters, while the interaction between factors was not significant ($p > 0.05$) (Table 2).

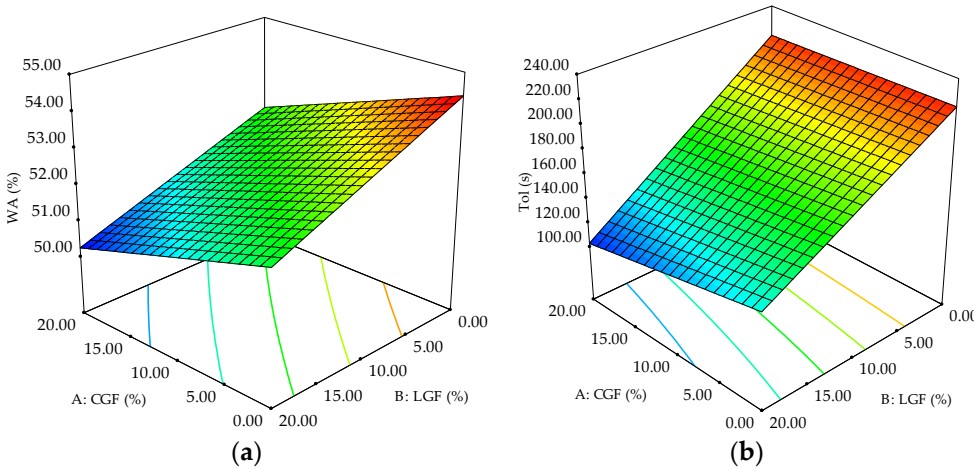

**Figure 2.** Response surface plot of (**a**) water absorption (WA) and (**b**) dough tolerance to kneading (Tol) as a function of chickpea germinated flour (CGF) and lupin germinated flour (LGF) addition.

The variations of dough consistency with CGF and LGF levels are presented in Figure 3. Since the model for D450 is first-order, the fitted response surface is a plane. The CGF factor did not affect significantly dough consistency after 250 and 450 s respectively ($p > 0.05$), while LGF addition led to the decrease of these parameters as the level raised (Figure 3). LGF showed significant influence ($p < 0.05$) on dough consistency parameters (Table 2), while the interaction between factors affected in a significant way only the consistency after 250 s.

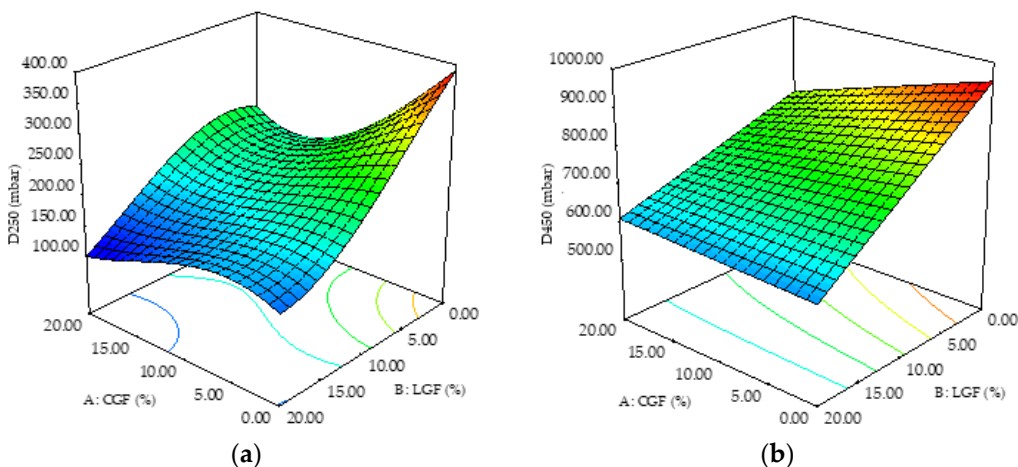

**Figure 3.** Response surface plot of (**a**) dough consistency after 250 s (D250) and (**b**) dough consistency after 450 s (D450) as a function of chickpea germinated flour (CGF) and lupin germinated flour (LGF) addition.

Dough rheological behavior during 3D-deformation could provide valuable information about handling proprieties. These characteristics are related to the quality and quantity of proteins, the incorporation of other ingredients in wheat flour causing their changes depending on the addition level. The nature surface plots depend on the signs and magnitudes of the coefficients in the model and are given by the mathematical model used. The hyperbolic surface plots presented in Figure 4 for elasticity and extensibility are characteristic for second-order polynomial model. Dough elasticity showed increases proportional to the amounts of CGF and LGF respectively, at levels between 10 and 15% (Figure 4). Both factors exhibited significant effects on dough elasticity ($p < 0.05$), while the interaction between them did not have a significant influence (Table 2). On the other hand, dough extensibility registered a decreasing trend as the amounts of CGF and LGF were higher (Figure 4), both factors and their interaction showing significant effects ($p < 0.05$) (Table 2).

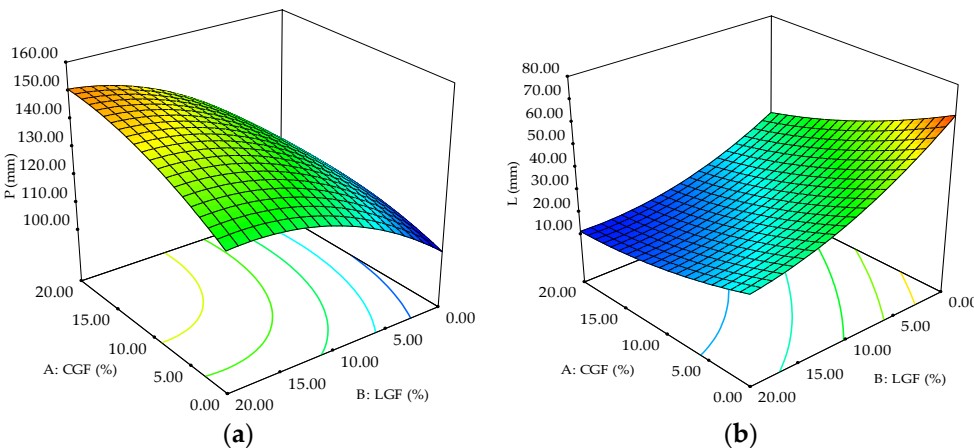

**Figure 4.** Response surface plot of (**a**) dough elasticity (P) and (**b**) dough extensibility (L) as a function of chickpea germinated flour (CGF) and lupin germinated flour (LGF) addition.

Alveograph energy and P/L were also significantly affected ($p < 0.05$) by CGF and LGF addition level, while their interaction influenced in a significant manner only P/L (Table 2). Since the models fitted to the data are first-order, the fitted response surfaces are plane. A decreasing trend of alveograph energy was observed as the amount of CGF and LGF raised, while opposite results were obtained for alveograph curve ratio variation (Figure 5).

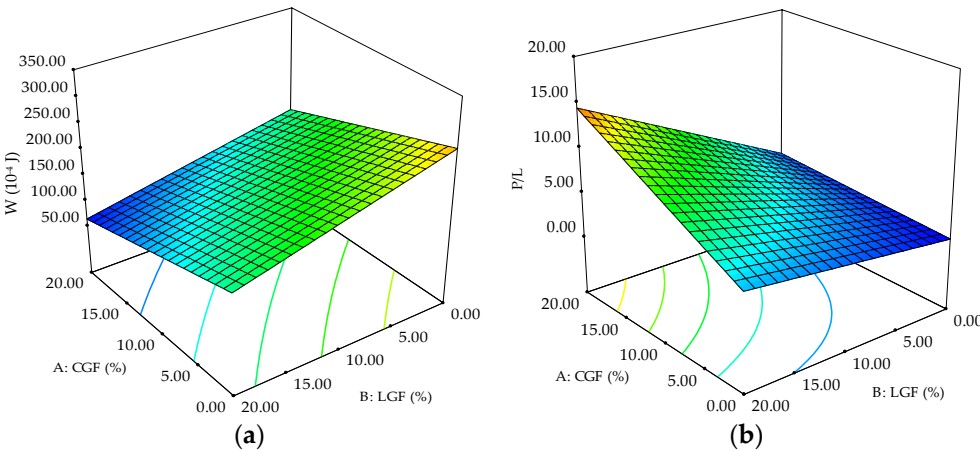

**Figure 5.** Response surface plot of (**a**) dough alveograph energy (W) and (**b**) alveograph curve ratio(P/L) as a function of chickpea germinated flour (CGF) and lupin germinated flour (LGF) addition.

### 3.4. Influence of CGF and LGF on Dough Rheological Behavior during Fermentation

Dough rheological properties during fermentation were changed by CGF and LGF incorporated in wheat flour. For the volume of the gas retained in the dough at the end of the test data variation the 2FI model was selected, a good fitting ($R^2 = 0.84$, $p < 0.05$) being obtained (Table 3). The results for maximum height of the gas release curve were successfully fitted to the quadratic model ($p < 0.05$) which explained 84% of data variation. Dough retention coefficient data variations were explained in a proportion of 60% by the cubic model which was significant ($p < 0.05$), while the data of total volume of $CO_2$ production were fitted to a modified cubic model ($R^2 = 0.87$, $p < 0.05$).

**Table 3.** ANOVA results of the mathematic models fitted for dough fermentation and fundamental rheological properties data.

| Factors | Parameters | | | | | | | | |
|---|---|---|---|---|---|---|---|---|---|
| | H′m (mL) | VT (mL) | VR (mL) | CR (%) | G′ (Pa) | G″ (Pa) | tan δ (adim.) | $T_i$ (°C) | $T_{max}$ (°C) |
| Constant | 62.13 | 1422.17 | 1122.04 | 80.75 | 34,481.30 | 12,858.60 | 0.3755 | 50.18 | 74.41 |
| A | −7.26 *** | −180.85 *** | −84.60 ** | 0.92 | −4956.17 | −1963.83 | −0.0095 | −0.63 *** | −1.33 ** |
| B | −10.60 *** | −322.04 *** | −154.36 *** | −5.46 | −18,140.36 ** | −6855.16 ** | 0.0097 | −0.98 *** | 0.13 |
| A × B | −3.53 | −37.56 | −35.52 | −0.50 | −1156.72 | −599.84 | −0.0153 | 0.39 * | −0.51 * |
| A² | 2.32 | 13.03 | | 0.17 | −4665.89 | −1548.06 | | −0.48 * | −0.38 |
| B² | −5.22 * | −61.60 | | −0.32 | −3866.63 | −1674.34 | | 0.05 | −0.10 |
| A²B | | 201.20 ** | | −6.70 * | 4647.66 | 1467.31 | | | −0.54 |
| AB² | | 63.94 | | −4.14 | 8509.14 * | 3016.00 * | | | −0.06 |
| A³ | | | | 4.41 | −1261.60 | −436.67 | | | 1.84 *** |
| B³ | | | | 12.43 ** | 8016.53 | 3493.20 | | | 0.16 |
| Model evaluation | | | | | | | | | |
| $R^2$ | 0.84 | 0.87 | 0.64 | 0.60 | 0.75 | 0.79 | 0.23 | 0.87 | 0.69 |
| *Adj.-$R^2$* | 0.79 | 0.82 | 0.59 | 0.47 | 0.60 | 0.67 | 0.12 | 0.83 | 0.55 |
| *p*-value | <0.0001 | <0.0001 | <0.0001 | 0.0493 | 0.0028 | 0.0008 | ns | <0.0001 | 0.0107 |

*** $p < 0.001$, ** $p < 0.01$, * $p < 0.05$, A: CGF, chickpea germinated flour addition (%); B: LGF, lupin germinated flour addition (%); $R^2$, *Adj.-$R^2$*, measures of model fit; H′m, maximum height of the gas release curve; VT, total volume of $CO_2$ production; VR, the volume of the gas retained in the dough at the end of the test; CR, retention coefficient; G′, G″, elastic and viscous modulus; tan δ, loss tangent; $T_i$, initial gelatinization temperature; $T_{max}$, maximum gelatinization temperature.

The maximum height of the gas release curve and the total volume of $CO_2$ production was significantly influenced ($p < 0.05$) by CGF and LGF factors (Table 3). The maximum height of the gas release curve decreased as the addition levels of CGF and LGF increased, a similar trend being observed in the case of total volume of $CO_2$ production as LGF amount raised, while CGF led to the increase of total volume of $CO_2$ production at levels up to 10%, then it decreased (Figure 6).

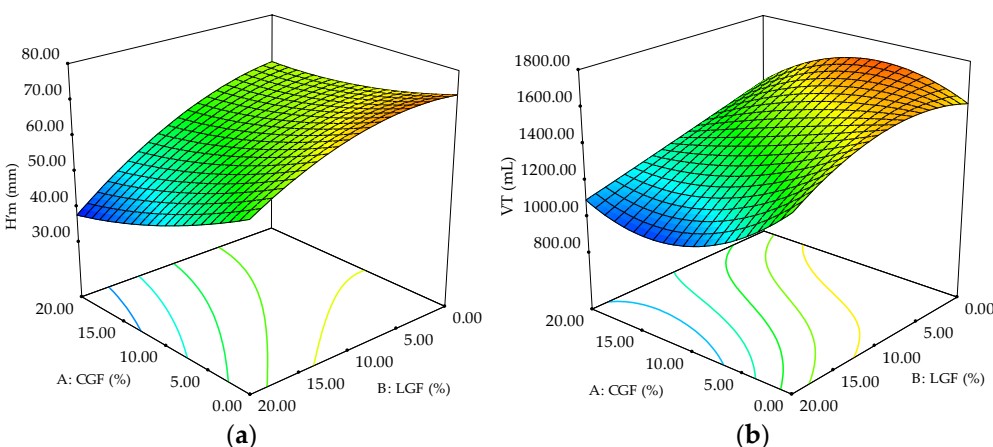

**Figure 6.** Response surface plot of (**a**) maximum height of the gas release curve (H'm) and (**b**) to-tal volume of $CO_2$ production (VT) as a function of chickpea germinated flour (CGF) and lupin germinated flour (LGF) addition.

Since the data for volume of the gas retained in the dough at the end of the test were fitted to a first-order mathematical model, the fitted response surface is a plane. The volume of the gas retained in the dough at the end of the test significantly decreased ($p < 0.05$) as the addition level of CGF and LGF raised (Figure 7), while the gas retention coefficient was significantly affected only by the interaction between the quadratic term of CGF and LGF and by the cubic term of LGF (Table 3).

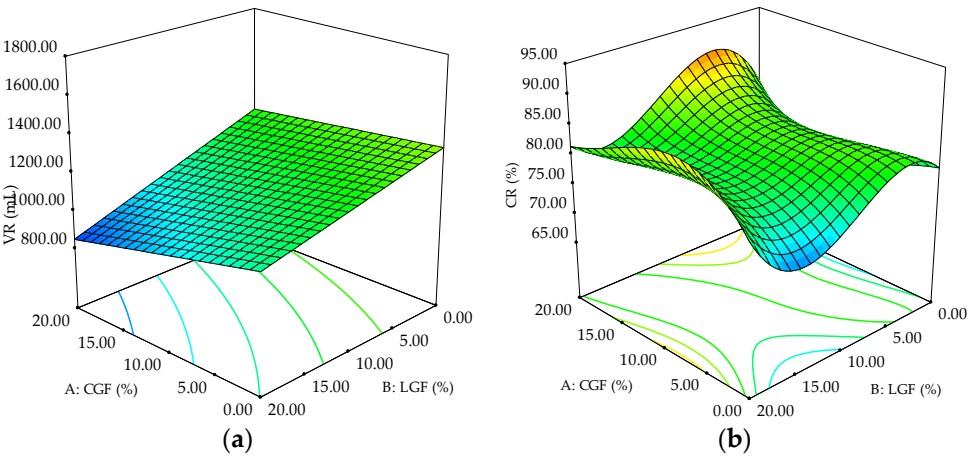

**Figure 7.** Response surface plot of (**a**) volume of the gas retained in the dough at the end of the test (VR) and (**b**) gas retention coefficient (CR) as a function of chickpea germinated flour (CGF) and lupin germinated flour (LGF) addition.

### 3.5. Influence of CGF and LGF on Dough Fundamental Rheological Properties

Dough visco-elastic behavior was influenced by CGF and LGF addition in wheat flour. For the loss tangent data variation, the 2FI model was non-significant ($p = 0.12$) (Table 3). The results for initial gelatinization temperature ($T_i$) were successfully fitted to the quadratic model ($p < 0.05$) which explained 87% of data variation. Dough elastic modulus, viscous modulus, and maximum gelatinization temperature data variations were explained in a proportion of 69–79% by the cubic models which were significant ($p < 0.05$).

The surface plots obtained are characteristic to the cubic models used for the visco-elastic moduli data and are given by the interaction and pure terms coefficients. The elastic and viscous moduli were significantly influenced ($p < 0.05$) only by the LGF factor and by its quadratic term interaction with CGF (Table 3). The elastic and viscous moduli increased

at CGF addition levels up to 10%, then the values decreased, while LGF caused the raise of these values at levels up to 5%, then they were reduced (Figure 8).

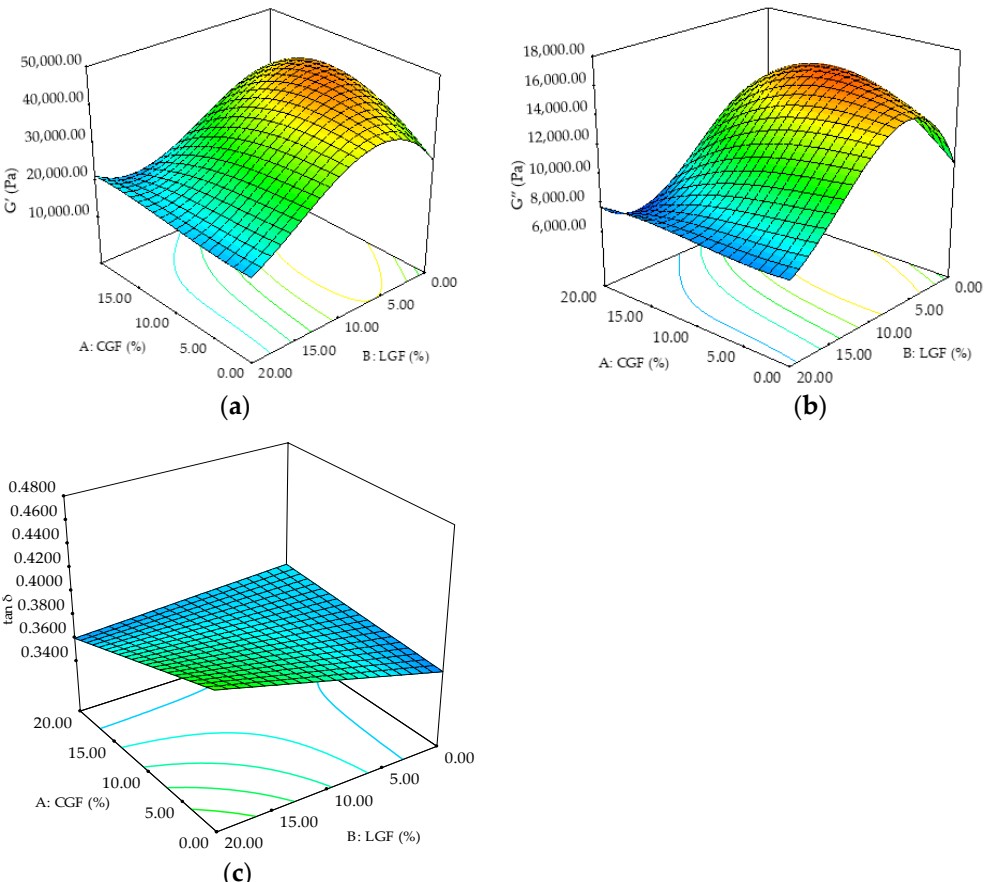

**Figure 8.** Response surface plot of (**a**) dough elastic modulus (G′), (**b**) dough viscous modulus (G″) and (**b**) loss tangent (tan δ) as a function of chickpea germinated flour (CGF) and lupin germinated flour (LGF) addition.

Dough behavior during heating can offer valuable information about starch gelatinization and protein denaturation during baking. The initial gelatinization temperature was significantly influenced ($p < 0.05$) by both factors and their interaction, a reduction being observed as CGF and LGF were raised (Figure 9). The maximum gelatinization temperature was significantly affected by CGF addition, an irregular trend being obtained, while the LGF factor showed a non-significant effect (Table 3).

### 3.6. Optimal Combination of CGF and LGF Added in Wheat Flour

The properties of the optimal sample compared to the control are listed in Table 4. The optimal formulation from a rheological point of view was obtained by incorporating 8.57% CGF and 5.31% LGF in wheat flour (Table 4). The objective of this study (to decrease the falling number close to the recommended range of $250 \pm 25$ s) was achieved, and a significantly lower value ($p < 0.05$) was obtained for the optimal sample compared to the control. The optimal sample presented lower water absorption, tolerance to kneading, dough consistency, extensibility, the volume of the gas retained in the dough at the end of the test, and initial gelatinization temperature compared to the control. The values for dough elasticity, alveograph curve ratio, the maximum height of the gas release curve, total volume of $CO_2$ production, gas retention coefficient, visco-elastic moduli, and maximum gelatinization temperature were higher for the optimal sample compared to the control.

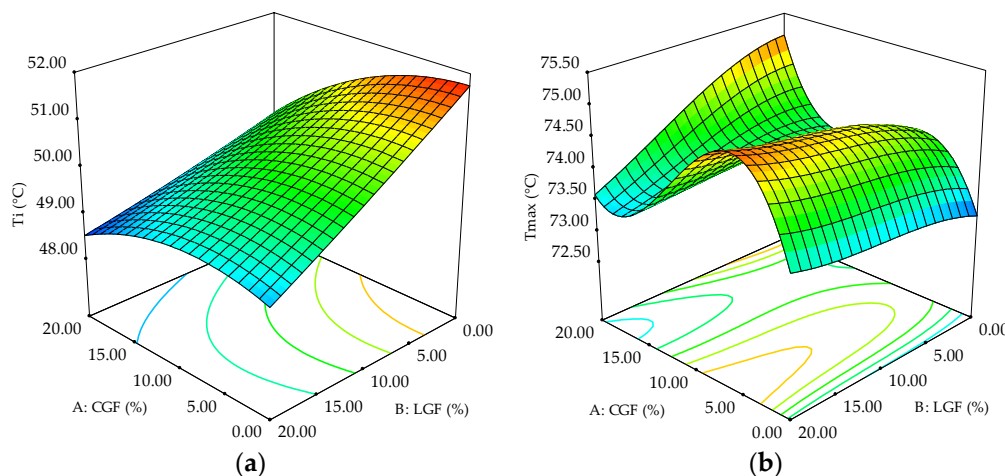

**Figure 9.** Response surface plot of (**a**) initial gelatinization temperature ($T_i$) and (**b**) maximum gelatinization temperature ($T_{max}$) as a function of chickpea germinated flour (CGF) and lupin germinated flour (LGF) addition.

**Table 4.** Characteristics of the optimal and control samples.

| Variable | Optimal Sample | Control |
|---|---|---|
| A: CGF (%) | 8.57 | 0.00 |
| B: LGF (%) | 5.31 | 0.00 |
| FN (s) | 283 [b] | 350 [a] |
| WA (%) | 52.9 [b] | 54.3 [a] |
| Tol (s) | 191.49 [b] | 214.00 [a] |
| D250 (mbar) | 238.55 [b] | 394.00 [a] |
| D450 (mbar) | 806.45 [b] | 943.00 [a] |
| P (mm) | 124.55 [a] | 104.00 [b] |
| L (mm) | 36.15 [b] | 72.00 [a] |
| W ($10^{-4}$ J) | 183.65 [b] | 301.00 [a] |
| P/L (adim.) | 3.78 [a] | 1.43 [b] |
| H′m (mL) | 66.80 [a] | 65.90 [b] |
| VT (mL) | 1579.31 [a] | 1532.00 [b] |
| VR (mL) | 1204.14 [b] | 1228.00 [a] |
| CR (%) | 81.97 [a] | 80.10 [b] |
| G′ (Pa) | 41,538.70 [a] | 29,290.00 [b] |
| G″ (Pa) | 15,446.25 [a] | 10,780.00 [b] |
| $T_i$ (°C) | 50.75 [b] | 51.90 [a] |
| $T_{max}$ (°C) | 74.46 [a] | 73.40 [b] |

A: CGF, chickpea germinated flour addition (%); B: LGF, lupin germinated flour addition (%); H′m, maximum height of the gas release curve; VT, total volume of $CO_2$ production; VR, the volume of the gas retained in the dough at the end of the test; CR, retention coefficient; G′, elastic modulus; G″, viscous modulus; $T_i$, initial gelatinization temperature; $T_{max}$, maximum gelatinization temperature; FN, falling number; WA, water absorption; Tol, tolerance to kneading; D250/D450, dough consistency after 250 and 450 s, respectively; P, dough elasticity; L, dough extensibility; W, alveograph energy; P/L, alveograph curve ratio. [a,b]—values signed by the same letter are not statistically different ($p > 0.05$).

In general, the data obtained indicate the fact that the addition of CGF in a higher amount and LGF in a lower one in wheat flour with low alpha-amylase activity may improve the technological behavior of dough during bread making, finally leading to bakery products of a higher quality. Therefore, the CGF and LGF combination may be considered as a bakery-improver for wheat flour which in addition of enhancing the nutritional characteristics of bakery products, it also improves their quality from the technological point of view. Of course, the addition levels of germinated flours may be adjusted in function of the wheat flour quality which is used as base for bread making. According to our data, the germinated flours used are not recommended for wheat flour

with low extensibility and higher tenacity, but are recommended for those with low alpha-amylase activity.

### 3.7. Relations between Variables

The Pearson's correlation coefficients between variables are presented in Table 5. Significant correlations ($p < 0.05$) were obtained between falling number and all the rheological parameters determined during mixing, 3D-deformation and fermentation and visco-elastic moduli. Water absorption was strongly correlated ($p < 0.05$) with the alveograph, consistograph, rheofermentometer (except retention coefficient when the correlation was weaker) and dynamic rheological parameters (except loss tangent and maximum gelatinization temperature). Significant correlations ($p < 0.05$) were obtained between the empirical rheological properties (water absorption, tolerance to kneading, consistencies, elasticity, extensibility, alveograph energy, P/L) and the dynamic elastic and viscous moduli. The alveograph parameters were significantly correlated with dough properties during fermentation, except dough elasticity which was not correlated with retention coefficient. Significant strong positive correlations ($p < 0.05$, $r > 0.63$) were observed between visco-elastic moduli and dough properties during fermentation, except retention coefficient. The initial gelatinization temperature was significantly correlated ($p < 0.05$, $-0.72 < r < 0.85$) with all the rheological parameters determined during mixing, 3D-deformation and fermentation and visco-elastic moduli.

**Table 5.** Pearson's correlation coefficients.

| Variables | FN | WA | Tol | D250 | D450 | P | L | W | P/L | G' | G'' | tan δ | $T_i$ | $T_{max}$ | H'm | VT | VR | CR |
|---|---|---|---|---|---|---|---|---|---|---|---|---|---|---|---|---|---|---|
| FN | **1.00** | **0.97** | **0.77** | **0.83** | **0.74** | **−0.84** | **0.96** | **0.93** | **−0.85** | **0.61** | **0.63** | −0.09 | **0.85** | −0.14 | **0.85** | **0.87** | **0.77** | −0.39 |
| WA | | **1.00** | **0.77** | **0.83** | **0.72** | **−0.78** | **0.93** | **0.93** | **−0.88** | **0.59** | **0.60** | −0.05 | **0.85** | −0.06 | **0.87** | **0.86** | **0.79** | −0.35 |
| Tol | | | **1.00** | **0.78** | **0.86** | **−0.72** | **0.73** | **0.69** | **−0.78** | **0.73** | **0.73** | −0.24 | **0.78** | 0.04 | **0.76** | **0.76** | **0.73** | −0.25 |
| D250 | | | | **1.00** | **0.87** | **−0.72** | **0.84** | **0.80** | **−0.70** | **0.44** | **0.45** | −0.16 | **0.79** | −0.17 | **0.70** | **0.66** | **0.72** | −0.07 |
| D450 | | | | | **1.00** | **−0.66** | **0.75** | **0.66** | **−0.64** | **0.53** | **0.53** | −0.24 | **0.77** | −0.12 | **0.65** | **0.67** | **0.67** | −0.18 |
| P | | | | | | **1.00** | **−0.84** | **−0.74** | **0.83** | **−0.53** | **−0.55** | 0.02 | **−0.67** | −0.09 | **−0.79** | **−0.78** | **−0.74** | 0.25 |
| L | | | | | | | **1.00** | **0.95** | **−0.83** | **0.51** | **0.53** | −0.04 | **0.83** | −0.14 | **0.77** | **0.79** | **0.72** | −0.33 |
| W | | | | | | | | **1.00** | **−0.86** | **0.48** | **0.51** | 0.06 | **0.82** | −0.12 | **0.75** | **0.75** | **0.68** | −0.32 |
| P/L | | | | | | | | | **1.00** | **−0.58** | **−0.62** | −0.08 | **−0.72** | −0.17 | **−0.86** | **−0.82** | **−0.77** | 0.29 |
| G' | | | | | | | | | | **1.00** | **0.99** | −0.44 | **0.66** | −0.02 | **0.64** | **0.68** | **0.63** | −0.25 |
| G'' | | | | | | | | | | | **1.00** | −0.29 | **0.67** | −0.01 | **0.70** | **0.71** | **0.67** | −0.25 |
| tan δ | | | | | | | | | | | | **1.00** | −0.19 | 0.15 | 0.12 | 0.00 | 0.03 | 0.03 |
| $T_i$ | | | | | | | | | | | | | **1.00** | −0.20 | **0.68** | **0.72** | **0.64** | −0.32 |
| $T_{max}$ | | | | | | | | | | | | | | **1.00** | 0.01 | −0.03 | −0.01 | 0.05 |
| H'm | | | | | | | | | | | | | | | **1.00** | **0.95** | **0.92** | −0.32 |
| VT | | | | | | | | | | | | | | | | **1.00** | **0.88** | −0.49 |
| VR | | | | | | | | | | | | | | | | | **1.00** | −0.02 |
| CR | | | | | | | | | | | | | | | | | | **1.00** |

Values in bold are significant at $p < 0.05$. FN, falling number; WA, water absorption; Tol, tolerance to kneading; D250/D450, dough consistency after 250 and 450 s, respectively; P, dough elasticity; L, dough extensibility; W, alveograph energy; P/L, alveograph curve ratio; H'm, maximum height of the gas release curve; VT, total volume of $CO_2$ production; VR, the volume of the gas retained in the dough at the end of the test; CR, retention coefficient; G', G'', elastic and viscous modulus; tan δ, loss tangent; $T_i$, initial gelatinization temperature; $T_{max}$, maximum gelatinization temperature.

The results of the principal component analysis (PCA) plot is presented in Figure 10. The first principal component (PC1) explained 64.59% of data variation, while the second one (PC2) explained 9.11% of the variance. Maximum gelatinization temperature and retention coefficient have a small contribution to the data variation, as it is suggested by their position on the graphic, close to the center. Reversely, closeness of single parameters as for example G' and G'', D250 and D450, $T_i$ and Tol, VT and FN, VR and FN, WA and FN, WA and L, L and H'm, P and P/L confirms a tight pair correlation. The PC1 was associated with retention coefficient, dough elasticity, P/L ratio, tolerance to kneading, dough consistencies D250, D450, falling number, volumes of gas, water absorption, alveograph energy and dough extensibility, while PC2 was associated with maximum gelatinization temperature and loss tangent.

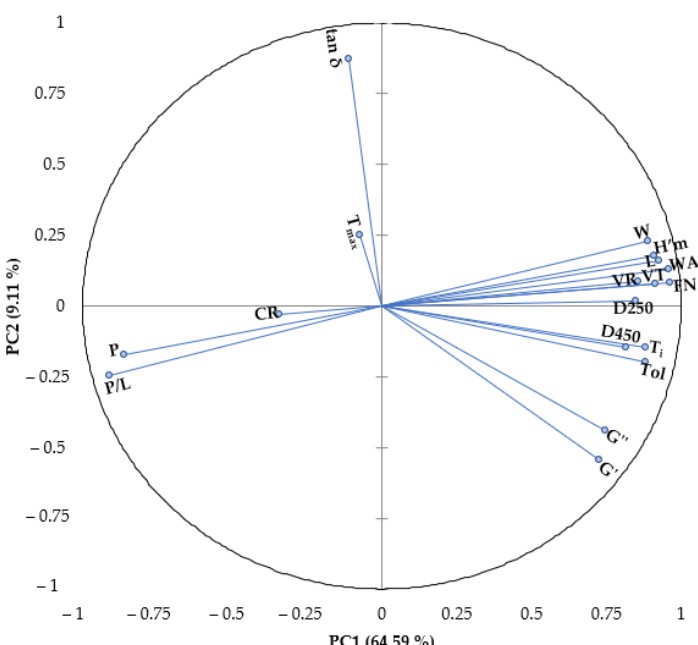

**Figure 10.** Principal component analysis plot of variables: FN, falling number; WA, water absorption; Tol, tolerance to kneading; D250/D450, dough consistency after 250 and 450 s, respectively; P, dough elasticity; L, dough extensibility; W, alveograph energy; P/L, alveograph curve ratio; H′m, maximum height of the gas release curve; VT, total volume of $CO_2$ production; VR, the volume of the gas retained in the dough at the end of the test; CR, retention coefficient; G′, G″, elastic and viscous modulus; tan δ, loss tangent; $T_i$, initial gelatinization temperature; $T_{max}$, maximum gelatinization temperature.

### 3.8. Characterization of Optimal and Control Sample Dough Microstructure

Figure 11 shows the overlay of the images obtained by fluorescence microscopic methodology using as fluorochroms rhodamine B and fluoresceine. In a dough system, starch and proteins are non-covalently labelled with fluoresceine and rhodamine B. Fluoresceine stains starch in green whereas rhodamine B stains protein in red [26]. As it may be seen from the both images obtained the starch matrix is the major compound from the dough system. When CGF and LGF were added in wheat flour the red area increased due to the increasing amount of protein concentration in the dough system. This is a consequence of the rhodamine B accumulation in dough due to it hydrophobic affinities in the protein phase [32].

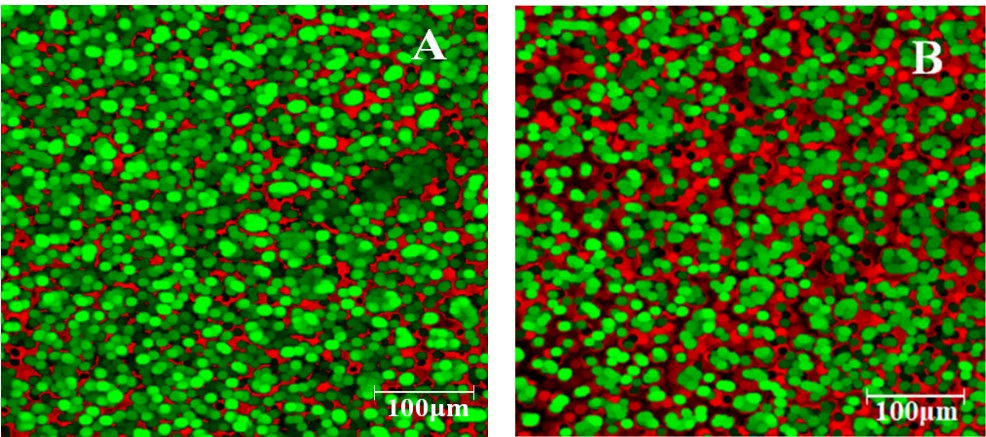

**Figure 11.** Dough microstructure of (**A**) wheat control sample (**B**) optimal combination of CGF, LGF, and wheat flour.

## 4. Discussion

Germinated legumes flour is a suitable alternative for chemical additives to improve wheat flour quality and consequently to obtain bread with higher nutritional benefits. The addition of CGF and LGF in white wheat flour resulted in significant changes in falling number values and rheological properties of dough.

Flour alpha-amylase activity was improved proportionally to the amount of CGF and LGF, as indicated by falling number values decrease. These results could be due to the intake of endogenous amylases from lupin and chickpeas which may break starch molecules during cooking in the falling number test, similar results being reported by Setia et al. [33]. Hernandez-Aguilar et al. [34] obtained a reduction of the falling number of wheat flour when more than 10% germinated lentil was incorporated.

The water absorption of dough decreases with CGF and LGF addition level increase and this could be related to the enzymatic activity of the germ on the starch molecules hydrolyzing them to dextrins which present low water binding capacity [35]. Furthermore, the proteins and fibers from CGF and LGF could contribute to wheat flour water absorption changes [11]. Dough kneading tolerance decreased as the amount of SGF and LGF increased, which was in agreement with the trend reported by Eissa et al. [36] for dough with germinated kidney pea flour. These results can be related to the lowering of glucose oxidase content which is known to possess a strengthening effect on the dough and/or to the raising of protease activity which hydrolyzes the gluten chains, resulting in a reduction of the overall dough strength [37]. The addition of LGF determined the decrease of dough consistency parameters which can be due to the pentosans found in lupin [38,39] that can interfere with gluten during mixing [40]. Furthermore, the increased enzymatic activities of flour mixes could have played also an essential role in dough rheological behavior during mixing. Significant positive correlations were observed between water absorption and dough consistency and tolerance to kneading ($p < 0.05$, $r > 0.72$).

The alveograph parameters of wheat dough were also significantly changed by wheat supplementation with CGF and LGF. Dough elasticity increased proportionally to the levels of CGF and LGF higher than 10%, most probably due to the content of cellulose present CGF and LGF [38], which can form strong interactions with wheat flour proteins [11]. Dough extensibility and alveograph energy decreased, while alveograph curve ration increased as the levels of CGF and LGF were higher. Germinated lupin and chickpea are sources of proteins and fibers which may interfere with wheat flour components [22,23]. The interactions of CGF and LGF chemical compounds with the gluten-starch matrix components seem to restrict the extensibility [20]. Our results are in agreement with those previously reported in the literature [20,21]. Weakening effects of CGF and LGF respectively on the mixed dough were obtained, taking into account that the alveograph energy values decreased as the addition level was higher. One reason for dough alveograph energy reduction could derive from legumes protein content which substitutes gluten proteins, causing a dilution effect and consequently a weakening of the dough [21]. The ratio of glutenin and gliadin content is crucial for dough optimal gluten network development, fact supported by the decrease of alveograph energy with CGF addition even if it contains glutenin [41]. These statements are supported also by the strong correlations of water absorption with the alveograph parameters ($p < 0.05$).

The incorporation of CGF and LGF in wheat flour resulted in a decrease in the maximum height of the gas release curve and volume gas. These results could be due to the weakened interactions between wheat and legumes proteins which determined a dough with a lower ability to maintain the gas produced during fermentation [42]. The reduction of the gas production and retention capacity of dough can be linked probably to the insoluble dietary fiber of CGF and LGF which produce the gluten dilution and/or may interact with gluten proteins [11], a fact supported also by the correlations obtained with water absorption capacity. In addition, the increase of enzymes activity during germination could exert a negative influence on gas retention which may be related to the weakening of dough caused by starch hydrolysis which is led to higher dough permeability [43]. Furthermore,

the ability of the dough to enclose air could have been altered by the protease hydrolyses peptide linkages which might cause partial destruction of the protein network [43].

The elastic and viscous moduli showed increasing trends proportional to the addition level of CGF and LGF. Dough elasticity raise can be caused by the increase of thiol group or a sulfhydryl group (SH) number, which determine dough oxidation through the mechanical action, the transformation of SH-bonds in disulfide bond (SS-bond), this newly formed SS-bond contributing to the increase in dough elasticity [21]. The increased amounts of proteins and polysaccharides, especially dietary fibers from CGF and LGF, may interfere with proper hydration and development of the gluten matrix, decreasing gluten plasticity [20]. Collar and Angioloni [42] stated that the addition of legume flours in wheat flour resulted in a significant effect on dough fundamental rheological characteristics which are mainly dependent on flour biopolymers, the rheological behavior of dough being dependent on the quantity and nature of the legume flour added. The initial gelatinization temperature was reduced as the addition level of CGF and LGF raised, while the maximum gelatinization temperature was influenced by CGF irregularly. The lipids and proteins from CGF and LGF may affect dough gelatinization temperatures since they could form complexes with starch [44]. Germination could have impacted the starch structure of legumes in terms of amylose and amylopectin content, heterogeneity, and content of amylose-lipid complexes (similar observations having been made by Jan et al. [45] for germinated Chenopodium seeds).

The optimization of CGF and LGF addition in wheat flour revealed that 8.57% CGF and 5.31% LGF can be added to obtain the best rheological properties. The increase in alpha-amylase activity of wheat flour suggested by falling number lowering with 19% was achieved by incorporating CGF and LGF, which indicate the suitability of these legumes for low alpha-amylase activity wheat flour enhancement. Compared to the control, the optimal sample was characterized by lower water absorption with 3%, smaller tolerance to kneading with 11%, lower dough consistencies with 39% (D250) and 14% (D450), smaller extensibility with 50%, lower volume of the gas retained in the dough at the end of the test with 2%, and smaller initial gelatinization temperature with 2%. On contrast, the elasticity was 20% greater, the alveograph curve ratio increased with 164%, the maximum height of the gas release curve was higher with 1%, the total volume of $CO_2$ production raised with 3% the gas retention coefficient increased with 2%, the visco-elastic moduli were 42% greater, and the maximum gelatinization temperature increased with 1% for the optimal sample compared to the control. These changes could be due to the higher enzymatic activity, complexes formation and intake of compounds such as non-gluten proteins, fibers, and lipids from germinated legumes in the optimal sample compared to the control. Considering these parameters, the optimal sample supplemented with CGF and LGF can be characterized as strong since the dough elasticity was greater than 100 mm and the alveograph curve ratio was higher than 1.50 mm [46]. The alveograph energy is higher than 160 $10^{-4}$ J which classifies the optimal sample as "good quality flour" [46].

The images obtained for the two dough samples (control and optimal one) shown differences between red and green area. The addition of CGF and LGF enlarged the red area from the dough system due to the high amount of protein from the germinated legumes used compared to the wheat flour [46]. Therefore, through partial substitution of wheat flour by CGF and LGF addition the protein amount from the dough increased. However, both dough structures appear homogeneous and regular. Germinated legumes and wheat starch granules are completely enveloped by proteins which seem as a continuous phase fact that is critical for the dough workability. No black holes are incorporated into the dough meaning that both dough samples present good technological properties fact correlated with the fundamental and empirical properties for these samples.

## 5. Conclusions

Wheat flour alpha-amylase activity can be supported by incorporating germinated chickpea and lupin flours, extending the alternatives for chemical additives. The effects

of chickpea germinated flour (CGF) and lupin germinated flour (LGF) on falling number values of white wheat flour resulted in a decrease of its values, demonstrating a higher alpha-amylase activity of the flour mixes. The rheological behavior of dough during mixing in terms of water absorption, tolerance to kneading, and consistency showed a decrease of these parameters as the addition level of CGF and LGF increased. A similar trend was observed for the 3D-deformation properties, except for alveograph curve ratio and dough elasticity which increased proportionally to the number of germinated legumes added. Dough behavior during fermentation in terms of maximum height of the gas release curve, total volume of $CO_2$ production, and volume of the gas retained in the dough at the end of the test exhibited significant lowering of these parameters as the level of CGF and LGF raised; a similar trend was obtained for the dynamic rheological moduli and gelatinization temperatures. The optimization of factors in the function of the rheological characteristics considered underlined that an addition level of 8.57% CGF and 5.31% LGF in wheat flour would be recommended. The use of this combination of ingredients resulted in improved alpha-amylase activity indicated by the lower falling number values, lower water absorption, tolerance to kneading, dough consistency, extensibility, the volume of the gas retained in the dough at the end of the test, the initial gelatinization temperature compared to the control, higher dough elasticity, alveograph curve ratio, maximum height of the gas release curve, total volume of $CO_2$ production, gas retention coefficient, visco-elastic moduli, and maximum gelatinization temperature compared to the control. These results could be of real help for producers interested in technological amelioration of wheat flour by using natural alternatives and being aware of product diversification. Further research on CGF and LGF influence on bread quality characteristics would be needed to better understand the interactions and benefits of these ingredients. In advance, a remarkable change of bread quality could be expected based on the rheological tests carried out.

**Author Contributions:** D.A., M.U.-I., G.G.C. and S.M. contributed equally to the study design, collection of data, development of the sampling, analyses, interpretation of results, and preparation of the paper. All authors have read and agreed to the published version of the manuscript.

**Funding:** This work was supported by a grant from the Romanian Ministry of Education and Research, CNCS–UEFISCDI, project number PN-III-P1-1.1-TE-2019-0892, within PNCDI III.

**Institutional Review Board Statement:** Not applicable.

**Informed Consent Statement:** Not applicable.

**Data Availability Statement:** Not applicable.

**Acknowledgments:** This work was supported by a grant of the Romanian Ministry of Education and Research, CNCS—UEFISCDI, project number PN-III-P1-1.1-TE-2019-0892, within PNCDI III.

**Conflicts of Interest:** The authors declare no conflict of interest.

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
