# Peer review of "Germinated Chickpea and Lupin as Promising Ingredients for Breadmaking—Rheological Features"

_agronomy, doi:10.3390/agronomy11122588_

Round 1
Reviewer 1 Report
Dear authors,
you've selected interestant theme of research and carried out many rheological tests. Submitted manuscript is written in English of adequate level with a minimum typos. My main objection is related to the presentation concept of the results - owing to suppresed primary data, regression modelling and huge number of 3D-plots, the article has a minimal chance to be repeated into another lab as well as its potential application in praxis is minimal. Briefly, I may tagget your mpst as "science pour science" type (analogy to l'art pour l'art, you see...).
To improve presentation of the gained data, I strongly recommend correlation analysis together with the PCA method to show inter-relationship of recorded quality features.
The second objection is related to use of non-standard terminolgy for parameters of the alveograph test; usualy elasticity dominates over tenacity, alveograph (P/L) ratio over curve configuration ratio and the parameter W is commonly called alveograph energy, not baking strength.
Further minor suggestions are appointed along the text into the *.pdf file.
With regards
Reviewer

Author Response
You've selected interestant theme of research and carried out many rheological tests. Submitted manuscript is written in English of adequate level with a minimum typos. My main objection is related to the presentation concept of the results - owing to suppresed primary data, regression modelling and huge number of 3D-plots, the article has a minimal chance to be repeated into another lab as well as its potential application in praxis is minimal. Briefly, I may tagget your mpst as "science pour science" type (analogy to l'art pour l'art, you see...).
We would like to thank the reviewer for the close reading and for the proper recommendations.
To improve presentation of the gained data, I strongly recommend correlation analysis together with the PCA method to show inter-relationship of recorded quality features.
We performed a correlation analyisis and PCA, as recommended.
The second objection is related to use of non-standard terminolgy for parameters of the alveograph test; usualy elasticity dominates over tenacity, alveograph (P/L) ratio over curve configuration ratio and the parameter W is commonly called alveograph energy, not baking strength.
We replaced the terminology for alveograph test, as suggested.
Further minor suggestions are appointed along the text into the *.pdf file.
We tried to correct all the minor suggestions appointed in the pdf file.

Reviewer 2 Report
Dear Editor and Authors,
Manuscript is interesting and valuable. There is only few studies about using germinated legume flour in bread processing, some of them aren’t mentioned (see below) please complete. Material, method and result sections are well described and presented. Please complete discussion section by comparing proposed references.
Abstract
Regarding sentence “Improving the alpha-amylase activity of wheat flour represents an opportunity to valorize wheat grains of low baking quality. In this sense, germinated legumes can be used to increase the enzymatic activity, giving at the same time superior final product characteristics”
Abstract is not compatible with the introduction section. The nutritional value of the lupine flour should also be mentioned at the beginning of the abstract
Introduction
Introduction section is well described but please mention about other research on the influence of the partial replacement of corn-rice flour by chickpea and sweet lupine flours on the quality characteristics of gluten-free bread was analyzed by Yousif and Safaa (2014) but not mentioned in this study. Also Omari et al used germinated lupin flour for flat bread properties – these studies should be mentioned in introduction section and discussion. Please complete this before the aim of the study.
Material and method and also result sections are well described. Discussion section could be improved by comparing missed research (as proposed below).
Al Omari, D. Z., Abdul-Hussain, S. S., & Ajo, R. Y. (2016). Germinated lupin (Lupinus albus) flour improves Arabic flat bread properties. Quality Assurance and Safety of Crops and Foods, 8(1), 57–63. https://doi.org/10.3920/QAS2014.0441
Yousif, M., & Safaa, M. (2014). Supplementation of gluten-free bread with some germinated legumes flour. Journal of American Science, 10(3), 84–93. Retrieved from http://www.jofamericanscience.org/journals/am-sci/am1003/011_23450am100314_84_93.pdf
Author Response
Dear Editor and Authors,
Manuscript is interesting and valuable. There is only few studies about using germinated legume flour in bread processing, some of them aren’t mentioned (see below) please complete. Material, method and result sections are well described and presented. Please complete discussion section by comparing proposed references.
We would like to thank the reviewer for the appreciations and proper recommendations. We really appreciate your suggestions in order to improve the manuscript.
Abstract
Regarding sentence “Improving the alpha-amylase activity of wheat flour represents an opportunity to valorize wheat grains of low baking quality. In this sense, germinated legumes can be used to increase the enzymatic activity, giving at the same time superior final product characteristics”
Abstract is not compatible with the introduction section. The nutritional value of the lupine flour should also be mentioned at the beginning of the abstract
We included the nutritional value of germinated chickpea and lupin flours.
Introduction
Introduction section is well described but please mention about other research on the influence of the partial replacement of corn-rice flour by chickpea and sweet lupine flours on the quality characteristics of gluten-free bread was analyzed by Yousif and Safaa (2014) but not mentioned in this study. Also Omari et al used germinated lupin flour for flat bread properties – these studies should be mentioned in introduction section and discussion. Please complete this before the aim of the study.
Material and method and also result sections are well described. Discussion section could be improved by comparing missed research (as proposed below).
The Introduction and discussion sections were improved as suggested.
Al Omari, D. Z., Abdul-Hussain, S. S., & Ajo, R. Y. (2016). Germinated lupin (Lupinus albus) flour improves Arabic flat bread properties. Quality Assurance and Safety of Crops and Foods, 8(1), 57–63. https://doi.org/10.3920/QAS2014.0441
Yousif, M., & Safaa, M. (2014). Supplementation of gluten-free bread with some germinated legumes flour. Journal of American Science, 10(3), 84–93. Retrieved from http://www.jofamericanscience.org/journals/am-sci/am1003/011_23450am100314_84_93.pdf
We would like to thank to Reviewers for all their comments and suggestions which have helped us to correct our work and present it in a more acceptable form. All the modifications are marked with Track Changes function.

Round 2
Reviewer 1 Report
Dear authors,
I very much appreciate your efforts in terms of clearly improving the presentation and explaining the relationship between the measured data (thanks to the inclusion of correlation analysis and the PCA method). However, the description of the results of the latter statistical method could be somewhat broader, as indicated in note within the enclosed file.
Further, in the second version of your manuscript, many typos remained as a result of deleting or exchanging words. Also in § References, it is necessary to correct the typos.
Overall, I am satisfied with the current version of the article and only "Minor revision" without the second round of the peer-review process is requested now.
Best regards
Reviewer

Author Response
I very much appreciate your efforts in terms of clearly improving the presentation and explaining the relationship between the measured data (thanks to the inclusion of correlation analysis and the PCA method). However, the description of the results of the latter statistical method could be somewhat broader, as indicated in note within the enclosed file.
Further, in the second version of your manuscript, many typos remained as a result of deleting or exchanging words. Also in § References, it is necessary to correct the typos.
We would like to thank the reviewer for the appreciations and proper recommendations. We really appreciate your suggestions in order to improve the manuscript.
We corrected the errors indicated, we split Table 1 in Table 1a and Table 1b and also we completed the PCA description, according to the suggestion.
Overall, I am satisfied with the current version of the article and only "Minor revision" without the second round of the peer-review process is requested now.
All the modifications are marked with Track Changes function.
